# State of the Art in Lung Nodule Localization

**DOI:** 10.3390/jcm11216317

**Published:** 2022-10-26

**Authors:** Evan T. Alicuben, Renee L. Levesque, Syed F. Ashraf, Neil A. Christie, Omar Awais, Inderpal S. Sarkaria, Rajeev Dhupar

**Affiliations:** 1Department of Cardiothoracic Surgery, University of Pittsburgh, Pittsburgh, PA 15213, USA; 2Surgical Services Division, VA Pittsburgh Healthcare System, Pittsburgh, PA 15240, USA

**Keywords:** lung nodule, ground-glass opacity, localization, minimally invasive surgery

## Abstract

Lung nodule and ground-glass opacity localization for diagnostic and therapeutic purposes is often a challenge for thoracic surgeons. While there are several adjuncts and techniques in the surgeon’s armamentarium that can be helpful, accurate localization persists as a problem without a perfect solution. The last several decades have seen tremendous improvement in our ability to perform major operations with minimally invasive procedures and resulting lower morbidity. However, technological advances have not been as widely realized for lung nodule localization to complement minimally invasive surgery. This review describes the latest advances in lung nodule localization technology while also demonstrating that more efforts in this area are needed.

## 1. Introduction

The evaluation of lung nodules and ground-glass opacities (GGO) is a critical component of patient care in a variety of settings. Most commonly, a thoracic surgeon will be asked to evaluate these lesions to determine whether they represent early-stage lung cancer or metastatic cancer from a nonlung origin. The implications to treatment can be highly impactful as results from a surgical biopsy can result in a curative procedure (for primary lung cancer) or a drastic change in prognosis and therapy (for metastatic lesions). Unfortunately, transthoracic fine needle aspiration or endobronchial biopsy can be inadequate or unfeasible, depending on the nodule size, location, and implications for treatment. In addition, there are often targeted therapies or clinical trials that depend on a detailed analysis of these nodules for which nodule excision is required.

Most thoracic surgeons have relied on three fundamental methods for lung nodule and GGO localization during minimally invasive surgical procedures: (1) visual recognition or planned anatomic resection, (2) percutaneous marking (e.g., wire localization or fiducial placement), or (3) endobronchial marking (e.g., dye marking with navigational bronchoscopy). Each of these methods (alone or in combination) has advantages and often augments the thoracic surgeon’s ability to localize a nodule. However, there is a significant variation in technique by institution, as well as a plethora of papers describing minor modifications to these techniques, demonstrating the imperfect nature of the current practice [1,2,3,4,5,6].

The following review will examine the newest methods that have been used to improve upon the persistent difficulty with lung nodule and GGO localization. It will not describe modifications of the three fundamental methods above, but rather will feature new technologies in order to better understand the state of the art, future directions, and how we should focus current efforts.

## 2. Methods

A PubMed search was performed with the key words “lung”, “nodule”, and “localization”. Over 3000 articles resulted, and this was restricted to the last 10 years, which resulted in 1649 results. These were individually reviewed (ETA and RD), and the 24 relevant articles regarding new techniques were selected based on the topic. See the flow chart (Figure 1).

## 3. Review

### 3.1. Targeted Navigation

Implantable devices are being developed that can be placed near the lesion of interest (via bronchoscopy or CT-guided delivery), and then a probe can aid in the detection of the device during surgery to localize this area. A radiofrequency identification (RFID) marking system has been used at Kyoto University with some success [7]. A very small chip is placed in close proximity to the lesion of interest by CT-guided bronchoscopy, and then wedge resection is performed. In a series of 12 cases, they were 100% successful in resecting the lesion and marker, with a conclusion that the use of this technology is safe and precise.

Elucent Medical [8] is adapting a device used for localizing breast tumors (SmartClip^TM^) for delivery to the lung via CT-guided percutaneous placement (with a 17-gauge needle) or endobronchial delivery (through a 2 mm working channel). These clips will be detectable radiographically (fluoroscopy or ultrasound) or with their EnVisio^TM^ Navigation System, which is intended to be used during the operation with sterile probes placed on minimally invasive instruments. A real-time three-dimensional navigation screen aids in locating the clip with accurate distance, depth, and location from the probe. Up to three clips can be placed and tracked independently. This technology is in development, and clinical trials are likely.

### 3.2. Intraoperative Ultrasound and CT Scan

Ultrasound technology has been applied by numerous surgical subspecialties to localize lesions intraoperatively and has also been used in lung resection. In a study of 53 lesions in 44 patients, Kondo et al. was able to demonstrate successful identification of all ground-glass opacities seen on preoperative imaging [9]. Other studies have reported similar success with Khereba et al. localizing 43 of 46 nodules [10]. Ultrasound still has limitations, primarily being highly user dependent and having diminishing accuracy in patients with severe emphysema, where full atelectasis may not be achieved. Additionally, a dedicated ultrasound machine and thoracic probe are required.

Intraoperative CT scan can provide real-time imaging most comparable to preoperative imaging but is limited by the fact that most procedures are not performed in hybrid operating rooms and patients would need to be repositioned and transferred to a different bed to be scanned. The O-arm Surgical Imaging System (Medtronic Japan Co., Ltd., Tokyo, Japan) is a full rotation system that can provide three-dimensional cone beam imaging, generating CT images in real time. It can be performed with the patient lateral while maintaining the sterile field. In a case report, the technology was utilized following two wedge resections that initially did not yield the small nodule being resected [11]. After visualizing the nodule with the O-arm, a needle was placed to aid in successful wedge resection. In all, this was noted to prevent either conversion to thoracotomy and/or more radical lung resection. The major drawback is the requirement for a specialty OR.

### 3.3. 3-D Modeling

#### Virtual Models

With technological advances in both image acquisition and processing, the creation of sophisticated preoperative virtual models has been utilized. Thin-cut CT scan images are processed through software programs to create 3D representations of each lobe, including pleural contour, fissures, and bronchovascular structures. The nodule is then displayed within the model. This provides not only spatial orientation intraoperatively, but specific measurements can be made relative to known structures and replicated during minimally invasive procedures. In a study using MIMICS (Materialise Interactive Medical Image Control System, version 20.0, Materialise, Belgium), Zhang et al. reported on a successful localization of 117 nodules in 44 patients [12]. The mean nodule diameter was 7.7 mm, and 85.1% of the lesions were malignant. In patients requiring anatomic resection, it was noted that 3D reconstruction allowed for a detailed assessment of proximity to bronchovascular structures. The main drawback is the requirement for the specialized system and the reliance on basic anatomic principles that are similar without the 3D representation.

Lesions deep within the lung parenchyma remain the most challenging to locate intraoperatively and can even be difficult to find with direct palpation. In a multicenter, prospective study in Japan, Sato et al. reported on the results of their virtual-assisted lung mapping 2.0 (VAL-MAP 2.0) system for the resection of deep lung nodules [13]. Patients with nodules identified within the inner two-thirds of the parenchyma were included. Preoperative marking was performed consisting of bronchoscopic guided dye marks to the pleural surface that indicated adequate margins around the lesion. To determine the deep resection margin, microcoils were placed into distal bronchi. If a single bronchus was near the lesion, one coil was placed, but if the lesion was not near a single bronchus, multiple coils were placed in relevant bronchi around the lesion. Intraoperative fluoroscopy was then used to identify the coils and guide the deep resection margin. In their series, 65 lesions were resected in 64 patients. Successful resection defined as a margin the size of the nodule diameter or larger than 2 cm was achieved in 64 resections (98.5%). However, this process usually takes place over several patient visits and requires a separate bronchoscopic procedure, 3D modeling with in-house software, and intraoperative fluoroscopy.

### 3.4. Augmented Reality Assistance

Virtual reality immerses the user into a completely synthetic world without any possibility of seeing the real-world environment, except through computer-generated representations [14]. Augmented reality (AR), in comparison, uses virtual computer-generated content, such as a graphical overlay, and applies it to a real-world view [15]. An example of AR in lung nodule detection in clinical use today is in robotic-assisted bronchoscopy. The Monarch (Auris Health, Inc., Redwood City, California) system’s display shows the real-world bronchoscope camera output. A graphical overlay consisting of an arrow/arc on the edge of the camera output informs the user of the articulation of the scope and sheath [16]. Similarly, the Ion (Intuitive Surgical, Sunnyvale, California) system displays a graphical overlay of a virtual model of the nodule on a fluoroscopy X-ray image in real time, allowing the user to direct the scope towards that nodule [17]. While this is in commercial use for bronchoscopic procedures, it is not yet available for resectional procedures.

An AR headset is an apparatus that allows the user to view graphical overlays on the real environment through a hands-free, head-mounted screen or eyeglasses. The HoloLens (v1/v2, Microsoft, Redmond, Washington) has been used to display floating 3D models of patients’ lung anatomy and nodule location to enable preoperative surgical planning [18,19]. In the lab, researchers have utilized the HoloLens for CT-guided lung biopsy. Demonstrating this concept, Peng et al. and Li et al. performed CT scans of pigs or canines and segmented 3D models of lung nodules (virtual or implanted) from the scan [20,21]. A planning path of the needle from the skin to the lung nodule was computed, and this information transferred to the HoloLens unit. With the animal still on the CT scanner table, the user wears the HoloLens, where a graphical overlay guides the user to the skin entry point for the needle. As the needle is inserted through the skin, the user views a virtual model of the lung nodule and needle path, overlaid on the animal. This allows proceduralists to maintain visualization on the surgical area during critical portions of the procedure, while receiving feedback on the navigation of the needle position. What is yet to be determined is whether this type of system can be used in the OR, with the lung position changing and becoming deflated, or during an operation when portions of the lung are manipulated and resected.

### 3.5. Indocyanine Green (ICG)

The agent indocyanine green (ICG) has been used with near-infrared fluorescence imaging to help visualize lesions intraoperatively [22]. Most commonly, peritumoral lung parenchyma is injected with ICG via percutaneous or bronchoscopic approaches [23]. Intraoperative use of specialized cameras allows the visualization of these areas, targeting resection. Reports of its use have been associated with an 86%–100% localization rate [22,24]. Success appears to be related to the depth of the tumor to the lung edge. Alternatively, intravenous injection of ICG has also been explored. Given the increased vascularity of tumors, it is theoretically plausible that these areas should fluoresce brightly. In a study published by Okusanya et al., this strategy was found to be successful in locating 90% of lesions, but did have a high rate of uptake by nonmalignant nodules [25]. ICG is commercially available, but this type of localization requires preoperative tumor injection.

### 3.6. OTL-38

Conceptually, an ideal localizing technique would incorporate an agent specific to malignant lesions that can be given preoperatively without the need for invasive procedures. OTL-38 (On Target Laboratories, West Lafayette, IN) is a tissue-specific dye that binds a folate receptor that has been found to be highly expressed in greater than 80% of pulmonary adenocarcinomas [26]. Exposure to near-infrared light causes fluorescence. Predina et al. reported on the use of OTL-38 in the localization of ground-glass opacities [27]. When 21 lesions were imaged with near-infrared light, 15 were identified. When only VATS was utilized, 10 of the 21 lesions were located. Lesions smaller than 2 cm and less than 1.5 cm from the surface of the lung were most effectively found.

A multi-institutional phase 2 study published by Gangadharan et al. examined the utility of OTL-38 in influencing intraoperative surgical decision making [28]. In a cohort of 92 patients, information obtained from intraoperative molecular imaging resulted in a clinically significant event in 26% of the patients. These included the identification of 24 additional nodules, 9 of them found to be malignant. The use of imaging located 11 lesions that could not be found by the surgeon, and 8 positive margins were identified after specimen resection despite an anticipated negative margin from the surgeon. Results of a randomized phase 3 multi-institutional trial are pending. However, the benefits of a potentially tumor-specific localizing IV injection near the time of surgery is attractive. What is unclear is how specific this dye is to cancer and its utility in early-stage or nonmalignant nodules that still require resection.

### 3.7. Magnetic Injections

Injection of magnetic material provides another unique method of nodule localization. Garcia et al. described using an implantable surgical steel seed marker, initially developed for targeting nonpalpable breast lesions [29]. Using an 18-gauge needle, a 1 × 5 mm seed marker was placed in a lesion preoperatively in four patients. The Sentimag magnetic tracker (Endomagnetics Inc., Austin, Texas) was used intraoperatively with authors noting that nonmagnetic instruments need to be used to manipulate the lung when localizing. The nodules ranged from 5 to 13 mm, and all were successfully found.

Alternatively, the injection of magnetic fluid in lesions has been described. In an animal experiment reported by Zhang et al., a sodium alginate–Fe_3_O_4_ solution was mixed with a curing agent to create a solidifying localizer in 15 rabbits [30]. Animals were injected percutaneously with a 16-gauge needle into a hypothetical nodule. A magnet was then used intraoperatively to tent up the targeted area. There was no dispersion of the solution into the surrounding lung tissue, and no dissemination was found in other organs. The authors did note that there was fibrosis near the location of the magnetic gel. Additional studies will be needed to determine the effect this may have on pathologic diagnosis.

## 4. Discussion

This review examined the state of the art for nodule localization in the lung. What is clear is that there is no universally adopted standard, and there are many avenues that are being explored that will contribute to improved procedures. The fundamental concerns are that new techniques be readily adoptable across a variety of medical centers, that they be intuitive to thoracic surgeons, and that the need for specialty equipment or multidisciplinary coordination be reasonable. However, it is likely that the adoption of new techniques will require training and specialized equipment until there is a universal standard. Ideal would be a device that can be used by multiple surgeons (e.g., breast surgeons and thoracic surgeons) so that trainees will have significant experience prior to practice and equipment can be shared.

This review intended to introduce the newest technologies that are being applied to lung nodule localization. Unfortunately, because the scale of use remains limited to mostly single institutions and the state of research is case reports and case series, it is challenging to evaluate which are most likely to become widely adopted. We look forward to more robust data so that advantages and disadvantages can be ascertained.

While we believe that the availability of better technologies to localize nodules, such as robotic bronchoscopy and cone beam CT scanners, will greatly enhance our ability to find nodules, more trials and practical experience are needed to determine the direction of the field. Multidisciplinary cooperation toward the goal of improved localization will greatly facilitate the adoption of newer techniques, utilizing the best that thoracic surgeons, interventional pulmonologists, and radiologists can offer.

## Figures and Tables

**Figure 1 jcm-11-06317-f001:**
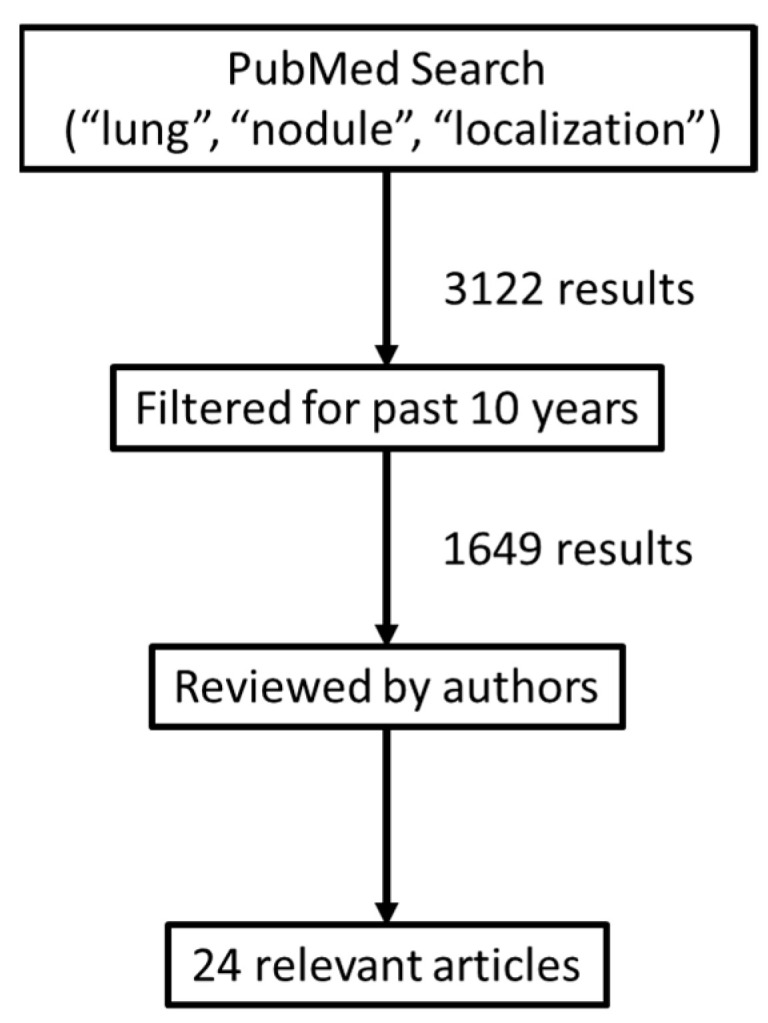
Flow chart of the selection process for article selection.

## Data Availability

Not applicable.

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
