# Peer review of "State of the Art in Lung Nodule Localization"

_jcm, 2022, doi:10.3390/jcm11216317_

Round 1
Reviewer 1 Report
I thank the editor to give mi the possibility to review this interesting paper. This is a narrative review regarding the state of art of the lung nodule localization. I have some questions to the authors to improve the article. First of all, also in case of a "narrative reivew" the methods of selection should be scientific. Please, add a flow-chart to explain how you select the papers cited. Then, in the last weeks, have been published some other review redarding this issue, I suggest the authors to cited its. Finally, I think that at the end of the review they should add a paragraph with the future perspective (robotic broncoscopy...).
Author Response
Reviewer 1
“Please, add a flow-chart to explain how you select the papers cited.”
Thank you for this suggestion. We have included a flow chart as Figure 1 and included in the methods our process of selection.
“Then, in the last weeks, have been published some other review redarding this issue, I suggest the authors to cited its.”
Thank you for this suggestion. We were unable to locate a review that is on the topic of the most up-to-date technologies and does not review traditional methods (fiducials, wires, or coil localization). We are happy to include the reference if the reviewer can be more specific regarding title and authors. We have included 4 new references of historic papers in the introduction (as suggested by a different reviewer).
“Finally, I think that at the end of the review they should add a paragraph with the future perspective (robotic broncoscopy...).”
Thank you for this suggestion. We have included a commentary on the future perspective at the end of the review.
“While we believe that the availability of better technologies to localize nodules, such as robotic bronchoscopy and cone beam CT scanners, will greatly enhance our ability to find nodules, more trials and practical experience are needed to determine the direction of the field. Multidisciplinary cooperation toward the goal of improved localization will greatly facilitate adoption of newer techniques, utilizing the best that thoracic surgeons, interventional pulmonologists, and radiologists can offer.”
Reviewer 2 Report
Dear Authors,
Congratulations on your manuscript. The topic chosen for this brief review is exciting and always relevant. The difficulty in intraoperative identification of small pulmonary nodules and GGO/mixed lesions is a stumbling block for the thoracic surgeon, especially in the era of minimally-invasive surgery and lung-sparing surgery.
The only suggestion concerns the choice of a bibliography. I would enhance the manuscript with less recent but critical studies to explain the stages of innovation that made the current targeting techniques possible.
Good job.
Author Response
Reviewer 2
“I would enhance the manuscript with less recent but critical studies to explain the stages of innovation that made the current targeting techniques possible.”
Thank you for this suggestion. We have included several references of less recent studies in the introduction.
Reviewer 3 Report
The authors stated that "State of the Art in Lung Nodule Localization". Although the topic of this research is relevant and important, the authors does not thoroughly analyze the research status in this field, meanwhile, does not fully discuss the advantages and disadvantages of various methods.
Author Response
Reviewer 3
“Although the topic of this research is relevant and important, the authors does not thoroughly analyze the research status in this field, meanwhile, does not fully discuss the advantages and disadvantages of various methods.”
Thank you for this comment. We have added additional information in the discussion to state that the research status is limited to case reports and no large scale trials or cohort studies due to the yet unproven nature of the technologies. We have also added that it is difficult to understand the advantages and disadvantages because of the small scale of use, but this will be critical ongoing.
This review intended to introduce the newest technologies that are being applied to lung nodule localization. Unfortunately, because the scale of use remains limited to mostly single institutions and the state of research is case reports and case series, it is challenging to evaluate which are most likely to become widely adopted. We look forward to more robust data so that advantages and disadvantages can be ascertained.
Round 2
Reviewer 1 Report
Dear authors, thank for the corrections. The paper is strongly improved. I suggest to add these recent references that have addressed your topic.
Chang, C.-J.; Lu, C.-H.; Gao, X.; Fang, H.-Y.; Chao, Y.-K. Safety and Efficacy of Cone-Beam Computed Tomography-Guided Lung Tumor Localization with a Near-Infrared Marker: A Retrospective Study of 175 Patients. Life 2022, 12, 494. https://doi.org/10.3390/life12040494
Tajè R, Gallina FT, Forcella D, Vallati GE, Cappelli F, Pierconti F, Visca P, Melis E and Facciolo F (2022) Fluorescence-guided lung nodule identification during minimally invasive lung resections. Front. Surg. 9:943829. doi: 10.3389/fsurg.2022.943829
Author Response
Thank you. The suggested references have been added in the introduction.
Reviewer 3 Report
The responses are reasonable.
Author Response
Thank you.